# AdvDINO: Domain-Adversarial Self-Supervised Representation Learning for Spatial Proteomics

**Stella Su**                                                    STELLAKTSU@GMAIL.COM
**Marc Harary**                                             MARCHARARY98@GMAIL.COM
**Scott J. Rodig**                                           SRODIG@BWH.HARVARD.EDU
**William Lotter**                                 LOTTERB@DS.DFCI.HARVARD.EDU
*Dana-Farber Cancer Institute, Brigham and Women's Hospital, & Harvard Medical School*

**Editors:** Accepted for publication at MIDL 2026

## Abstract

Self-supervised learning (SSL) has emerged as a powerful approach for learning visual representations without manual annotations. However, the robustness of standard SSL methods to domain shift—systematic differences across data sources—remains uncertain, posing an especially critical challenge in biomedical imaging where batch effects can obscure true biological signals. We present AdvDINO, a domain-adversarial SSL framework that integrates a gradient reversal layer into the DINOv2 architecture to promote domain-invariant feature learning. Applied to a real-world cohort of six-channel multiplex immunofluorescence (mIF) whole slide images from lung cancer patients, AdvDINO mitigates slide-specific biases to learn more robust and biologically meaningful representations than non-adversarial baselines. Across more than 5.46 million mIF image tiles, the model uncovers phenotype clusters with differing proteomic profiles and prognostic significance, and enables strong survival prediction performance via attention-based multiple instance learning. The improved robustness also extends to a breast cancer cohort. While demonstrated on mIF data, AdvDINO is broadly applicable to other medical imaging domains, where domain shift is a common challenge.

**Keywords:** self-supervised learning, domain shift, spatial proteomics, batch effects

## 1. Introduction

Self-supervised learning (SSL) has emerged as a powerful framework for visual representation learning without annotated data, enabling models such as DINOv2 (Oquab et al., 2024) to learn embeddings that facilitate diverse downstream applications. However, common contrastive and autoencoding objectives are prone to capturing domain-specific confounders, which undermines the robustness and generalizability of the resulting embeddings.

Improved methods for handling domain shift are especially needed for emerging biomedical imaging techniques like spatial proteomics, which was named Nature's Method of the Year in 2024. Spatial proteomics involves imaging many protein biomarkers simultaneously in the same tissue sample, resulting in rich, multi-channel images. These techniques are increasingly used in oncology to interrogate the composition and spatial structure of tumors (de Souza et al., 2024). After imaging a cohort of interest, a common workflow is to perform clustering to identify distinct subsets within the cohort, and then assess associations between these clusters and patient outcomes. While clustering is traditionally performed using hand-engineered features (e.g., counts of different cell types), SSL applied to the

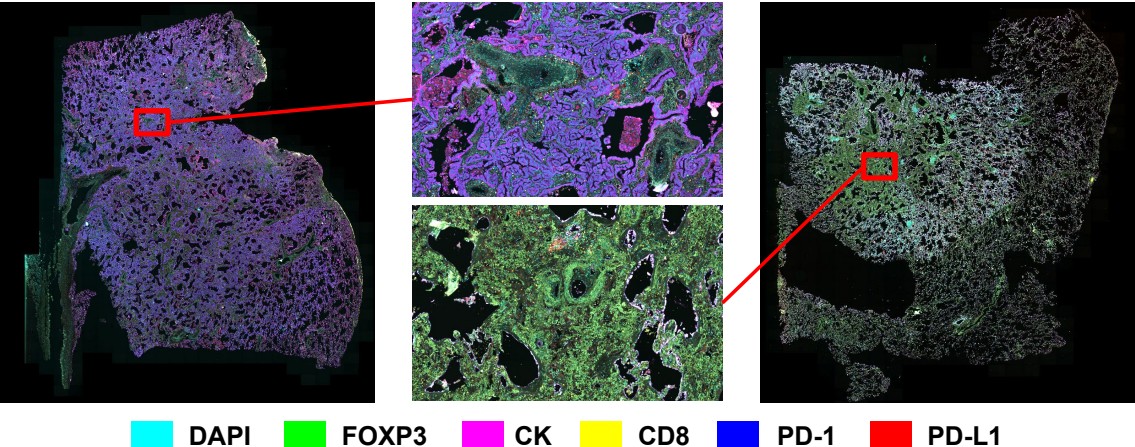

Figure 1: Domain shift in mIF WSIs. Two lung tumor samples show substantial variation in staining intensity and contrast due to batch effects, highlighting the challenge of domain shifts in biomedical imaging.

original images offers a promising data-driven solution. However, overall staining intensity, contrast, and signal distribution can vary dramatically between samples due to run-to-run variability (Figure 1), obscuring true biological signals.

To address this challenge, we propose AdvDINO, a domain-adversarial SSL framework that integrates a gradient-reversal-based domain discriminator into DINOv2. During pre-training, AdvDINO discourages the student encoder from capturing domain-identifiable features, while preserving semantic features that support downstream applications. While broadly applicable, we demonstrate the effectiveness of AdvDINO in spatial proteomics using a cohort of 435 lung cancer whole slide images (WSIs) with more than 5.46 million multi-channel image tiles. We perform unsupervised clustering on the tile-level embeddings learned by AdvDINO, and assess the biological and prognostic relevance of the resulting clusters. Towards direct prognostication, we apply attention-based multiple instance learning (ABMIL) to predict patient survival at the slide-level. We find that the model outperforms traditional hand-engineered metrics and that the domain-adversarial loss results in biologically-meaningful clusters that are shared across samples. We additionally assess the generalizability of the approach to a breast cancer cohort, identifying clusters associated with the aggressive triple negative breast cancer (TNBC) subtype. Together, these results highlight the robustness, interpretability, and downstream utility of AdvDINO.

## 2. Related Work and Background

**SSL for Visual Representation Learning:** SSL has shown strong performance in representation learning by leveraging pretext tasks that do not require manual labels. DINO (Caron et al., 2021) and its successor DINOv2 (Oquab et al., 2024) use a teacher–student framework with architectural asymmetry and knowledge distillation to learn rich image representations. DINOv2 has recently become the de facto standard for vision-based SSL and foundation model development in medical imaging (Chen et al., 2024; Xu et al., 2024;

Vorontsov et al., 2024; Pérez-García et al., 2025). However, the robustness of these approaches to domain shift remains uncertain. For instance, recent studies have shown that DINOv2-trained models in computational pathology are susceptible to batch effects (de Jong et al., 2025; Kömen et al., 2024; Mishra and Lotter, 2025).

**Domain Invariant Representation Learning:** Domain-Adversarial Neural Networks (DANN) (Ganin et al., 2016) introduced a gradient reversal layer to promote domain-invariant feature learning in supervised settings. While adversarial training has since been extended to various domain adaptation scenarios, many of these methods rely on task-specific supervision. Recent approaches like Style Augmentations for Self Supervised Learning (SASSL) (Rojas-Gomez et al., 2024) pursue domain invariance through data augmentation, using neural style transfer to improve feature robustness across domain variations. In contrast, AdvDINO integrates domain-adversarial learning directly into the self-supervised pretraining process.

**Spatial Proteomics Analysis:** Spatial proteomics techniques, especially those based on multiplex immunofluorescence (mIF), have been leveraged across many cancer types to quantify variations in the tumor microenvironment and identify features associated with patient outcomes (de Souza et al., 2024). In mIF, a selected set of proteins are imaged with fluorescent markers in a tissue sample. Commonly, this process is performed on sampled regions of a tissue section in the form of a tissue microarray (TMA), which represents a small proportion of the entire WSI. The set of markers used can vary assay-to-assay depending on the clinical context and goal. Traditionally, hand-engineered metrics such as the counts of specific cell types have been used to analyze the resulting images (Alessi et al., 2025). Several recent works have applied deep learning to mIF, including graph neural networks based on cellular neighborhoods within a TMA (Wu et al., 2022; Hoebel et al., 2026). Harary et al. (2024) developed an MIL approach for mIF WSIs that combines information across channels using off-the-shelf tile-level encoders developed for other image domains. In contrast, AdvDINO facilitates tile-level feature learning from mIF WSIs while mitigating batch effects.

## 3. Methods

### 3.1. AdvDINO Framework

**SSL via DINOv2:** The first component of AdvDINO is a student–teacher Vision Transformer (ViT; Dosovitskiy et al. 2021) pair trained using the self-distillation and masked image modeling (MIM) objectives introduced in DINOv2. Both the student and teacher networks are built on a ViT-L architecture that processes images by dividing them into non-overlapping 16×16 pixel patches. The ViT-L backbone comprises 24 stacked Transformer encoder layers, where each layer implements multi-head self-attention with 16 attention heads and processes token embeddings in a 1024-dimensional space.

**Domain-Adversarial Learning:** The second component of AdvDINO promotes domain-invariant representation learning through adversarial training, where the output of the student encoder serves as input to a domain discriminator (Figure 2). The internal architecture of the student ViT (the transformer blocks and attention mechanisms) remains the same

as the original DINOv2 backbone, but its weights are trained on both the self-supervised and adversarial objectives. The discriminator is trained to predict the domain label of an input image, while the student encoder is trained to produce features that hinder accurate domain classification. The encoder and discriminator are connected via a gradient reversal layer (GRL) to enable adversarial optimization (Ganin and Lempitsky, 2015). During the forward pass, the GRL behaves like the identity function by passing the input feature embedding from the encoder to the domain discriminator unchanged. However, during backpropagation, the GRL multiplies the gradients coming from the domain discriminator by $-1$ to reverse the direction of the gradient flow when updating the encoder, thus training the encoder to produce features that confuse the discriminator.

Each input image is associated with a domain label $d_j$ that reflects its origin, such as the slide, batch, or scanner. For each image $x_j$, multiple augmented views (i.e., image crops) $x_j^{(i)}$ are generated, where $j$ indexes the image and $i$ indexes the crop. The student encoder processes each crop $x_j^{(i)}$ to produce an embedding: $f_j^{(i)} = \text{Encoder}(x_j^{(i)})$, using the [CLS] token from the final transformer layer. The GRL then forwards the embedding to the domain discriminator: $\hat{d}_j^{(i)} = \text{DomainDiscriminator}(\text{GRL}(f_j^{(i)}))$. The domain discriminator is implemented as a lightweight multilayer perceptron (MLP) consisting of two fully connected layers with 256 and 128 units, respectively, each followed by a ReLU activation and a final softmax output layer that produces class probabilities for the domain classification task.

Following DINOv2's multi-crop strategy, each input image $x_j$ is augmented into two global crops and eight local crops, resulting in ten views $x_j^{(i)}$, where $i = 1, \ldots, 10$. Since these crops are all derived from the same input image, they share a common domain label $d_j$. Each crop is then passed individually through the student encoder, GRL, and domain discriminator. AdvDINO computes the adversarial loss independently for each crop $x_j^{(i)}$, ensuring that the discriminator is trained to classify the domain of each augmented view independently. The adversarial loss is computed using the standard cross-entropy objective: $\mathcal{L}_{\text{adv}} = \frac{1}{B \cdot N} \sum_{j=1}^{B} \sum_{i=1}^{N} \text{CE}(\hat{d}_j^{(i)}, d_j)$, where $B$ is the batch size, $N$ is the number of crops per image, and CE denotes the cross-entropy loss between predicted and true domain labels.

**Combined objective:** The AdvDINO training objective integrates the adversarial domain loss ($\mathcal{L}_{\text{adv}}$) with the original DINOv2 losses (MIM loss $\mathcal{L}_{\text{MIM}}$ and self-distillation loss $\mathcal{L}_{\text{distill}}$): $\mathcal{L}_{\text{total}} = \lambda_{\text{distill}} \mathcal{L}_{\text{distill}} + \lambda_{\text{MIM}} \mathcal{L}_{\text{MIM}} + \lambda_{\text{adv}} \mathcal{L}_{\text{adv}}$, where $\lambda_{\text{distill}}, \lambda_{\text{MIM}}, \lambda_{\text{adv}}$ are hyperparameters that control the relative contribution of each objective during training.

### 3.2. AdvDINO for mIF Image Analysis

**Adversarial Learning of Sample-Specific Staining:** While AdvDINO is flexible to the ways the domain labels are constructed, here we focus on a critical challenge for mIF analysis: learning representations that are robust to batch effects that result in slide-specific staining characteristics (Fig. 1). Therefore, we use the ID for each slide as the domain label for adversarial learning with a goal of learning robust tile-level embeddings. After SSL, we perform inference across all tiles using the ViT encoder trained with AdvDINO, producing 1024-dimensional embeddings per tile. These tile-level embeddings serve as inputs to two downstream tasks: (1) unsupervised clustering and (2) survival prediction via ABMIL.

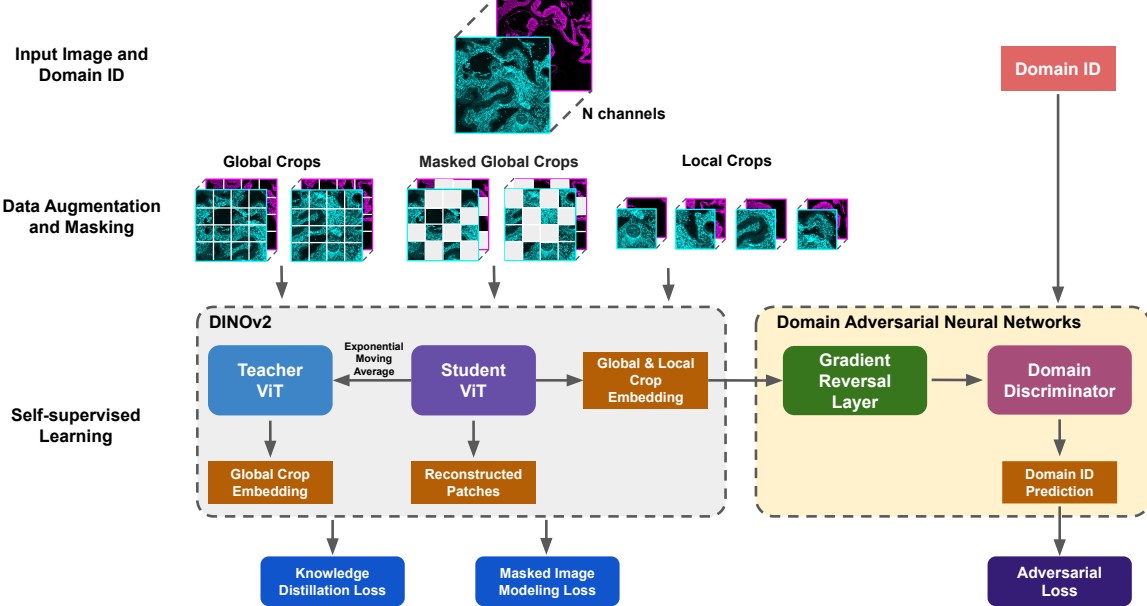

Figure 2: AdvDINO overview. AdvDINO integrates domain-adversarial learning into DI-NOv2 to learn domain-invariant representations from multi-channel image data.

**Unsupervised Clustering and Association Analysis:** Leiden clustering was performed on the learned embeddings (Traag et al., 2019), with details of the implementation included in the Appendix. To quantify the proteomic profile of the resulting clusters, we randomly sampled 2,000 tiles per cluster from the cohort and computed the mean pixel intensity for each of the mIF channels. These cluster-level averages were then z-scored across clusters to highlight their proteomic distinctions. To quantify the prognostic relevance of each cluster, we computed the abundance of each cluster per slide, defined as the proportion of tiles assigned to that cluster out of all tiles from the same slide. We then evaluated how cluster abundance correlated with patient survival by computing the concordance index (C-index) between cluster proportion and patient survival time.

**Survival Prediction with ABMIL:** For WSI-level survival prediction, we use ABMIL within a discrete survival model framework (Zadeh and Schmid, 2021). The ABMIL model is trained in a weakly-supervised fashion to predict patient overall survival using the tile embeddings. Details of the implementation are included in the Appendix.

### 3.3. Experimental Details

**Dataset:** The NSCLC cohort consists of 435 primary-site tumor samples from 414 patients that were prospectively profiled using the ImmunoProfile mIF assay from 2018-2022 at the Dana-Farber Cancer Institute (Alessi et al., 2025; Lindsay et al., 2025). The ImmunoProfile assay stains for four immune markers (CD8, FOXP3, PD-L1, PD-1), cytokeratin (CK) as a tumor marker, and DAPI as a counterstain for nucleus detection. Briefly, CD8 is a marker for cytotoxic T cells that can attack tumor cells (Raskov et al., 2021). FOXP3 is a marker

for regulatory T cells that can indicate immune suppression (Rudensky, 2011). PD-1 and PD-L1 are involved in immune inhibition and can be expressed by both immune and tumor cells (Han et al., 2020). The ImmunoProfile assay was performed on entire WSIs, in contrast to common TMA approaches.

For each sample, we obtained the mIF WSI, follow-up time, and survival status (deceased or censored). As an unselected clinical population, the dataset consists of tumors across different stages (306 or 70.5% low-stage, 125 or 28.8% high stage, 3 or 0.7% unknown stage) and different treatment regimes (97 or 22.4% receiving immunotherapy, 334 or 77.0% receiving treatment other than immunotherapies, 3 or 0.7% with unknown treatment), representative of a real-world clinical cohort. All data are de-identified. This research was IRB-approved by the Dana-Farber Cancer Institute under DFCI protocol 22–176.

**Input Preprocessing and Data Augmentation:** We applied AdvDINO to 256×256-pixel input tiles extracted from the mIF WSIs. The input tiles were extracted from foreground regions and contained 6 channels, one for each of the biomarkers in the assay. Further preprocessing details, including background subtraction and normalization, are described in the Appendix.

We applied data augmentations following the multi-crop strategy of DINOv2, with several adaptions for mIF. From each 256×256 tile, we generated two global crops of size 224×224, eight local crops of size 96×96, and two masked global crops. Each crop underwent random horizontal flips, random cropping, and channel-wise Gaussian blur. As an additional augmentation beyond the original DINOv2 pipeline, we included vertical flipping to increase invariance to orientation, which is applicable to pathology slides. Furthermore, we excluded solarization, which is not well-defined for non-RGB images.

**AdvDINO Training:** We trained AdvDINO on the entire 435 mIF WSI cohort, representing approximately 5.46 million tile images. We initialized both the student and teacher networks with pretrained weights from the UNI ViT-L/16 model trained on H&E tiles (Chen et al., 2024). Since the original patch embedding layer is designed for 3-channel (RGB) inputs, we adapted the pretrained weights by averaging across the RGB channels and replicating the result to initialize the 6-channel input layer. This approach preserves the pretrained spatial filters while enabling compatibility with the 6-marker mIF input format. Because the pretrained DINO heads are not available, we randomly initialized them. Other training hyperparameters generally followed those of Oquab et al. (2024) and Chen et al. (2024), as detailed in the Appendix. The new adversarial loss weight hyperparameter, $\lambda_{adv}$, was chosen such that the adversarial and SSL losses were approximately balanced during training, corresponding to a value of $\lambda_{adv}=50$. This strategy was implemented to facilitate domain invariance without compromising representation quality, and because DINOv2's high computational costs limit extensive hyperparameter optimization.

**Baselines:** We compare AdvDINO to a DINOv2 baseline with the same architecture and trained on the mIF cohort in the same fashion as AdvDINO, but without the domain discriminator. This comparison isolates the effects of the adversarial learning, as applied to a leading SSL algorithm in the field. For the ABMIL survival analysis, we additionally compare to a Cox proportional hazards (CoxPH) model fit using traditional cell density metrics. These metrics specifically consist of the intratumoral density of cells positive for

each immune biomarker (CD8, FOXP3, PD-1, PD-L1) and the PD-L1 tumor proportion score (TPS) (Alessi et al., 2025). Each of these metrics were percentile normalized across the cohort.

**Evaluation Metrics and Statistical Analysis:** We quantify the slide independence of the representations learned (i.e., the effectiveness of AdvDINO in mitigating domain shift) by computing the Adjusted Rand Index (ARI) between cluster assignments and slide ID labels for all tiles in the dataset. ARI measures agreement between two sets of labels while correcting for chance, where a value of 0 indicates chance concordance and 1 indicates perfect agreement. In our case, if each of the embedding clusters consist almost entirely of tiles from a single WSI, then the ARI would be high and it would suggest that the clusters reflect slide-specific batch effects rather than generalizable biological phenotypes. We also evaluate the prognostic relevance of the clusters by computing the C-index between the cluster abundance scores and patient survival. P-values for the abundance scores were calculated using the Wald test from a CoxPH regression and corrected for multiple comparisons using the Bonferroni method. For clusters with significant associations, an additional Cox regression stratified by cancer stage (binarized as high vs. low) was performed to evaluate whether the associations remained significant after adjusting for stage. For the ABMIL models, we perform five-fold cross-validation and also use the C-index as the evaluation metric (see Appendix for further details). This cross-validation scheme was also used for the baseline CoxPH model.

**Code Availability:** Code is available at https://github.com/lotterlab/advdino.

## 4. Results

We present results on (1) the domain robustness of AdvDINO representations, (2) the proteomic and prognostic characterization of clusters identified from the learned embeddings, and (3) survival prediction using ABMIL applied to the embeddings. The results are presented on a clinical cohort of 435 mIF WSIs from NSCLC patients, comprising over 5.46 million image tiles. The unsupervised clustering analysis was performed using the entire dataset, with five-fold cross validation performed for the supervised ABMIL analysis.

### 4.1. Domain Robustness Evaluation

We generated Uniform Manifold Approximation and Projection (UMAP) plots based on the embeddings from the AdvDINO and baseline DINOv2 models (Figure 3) to first visualize the structure of the tile embeddings. For visualization purposes only, the UMAPs were constructed from a random subset of 100 slides and 100 sampled tiles per slide. As shown in Figure 3A, the UMAP of tile embeddings from the baseline DINOv2 model, trained without adversarial loss, reveals pronounced slide-specific clustering. Each point corresponds to an individual tile and is colored by its slide of origin. Tiles from the same slide form tight, isolated groups, indicating that DINOv2 embeddings are heavily influenced by slide identity. A similar trend was observed for DINOv2 in UMAPs colored by cluster labels (Appendix Figure A1), where clusters obtained using the aforementioned pipeline reflected slide origin rather than generalizable biological variation. These results suggest that DINOv2, by

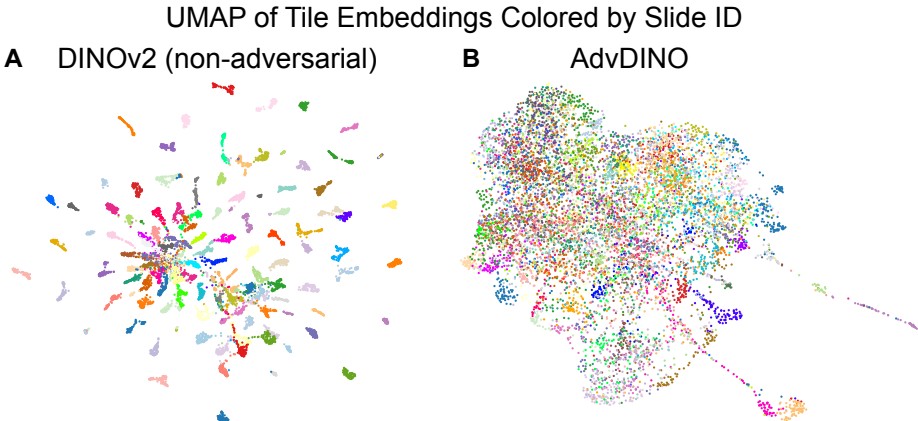

Figure 3: UMAP of tile embeddings from the baseline DINOv2 model (A) and AdvDINO (B), colored by slide ID.

default, overfits to slide-specific features, limiting its ability to generalize across slides and hindering the identification of biologically-meaningful clusters.

In contrast, the UMAP of AdvDINO embeddings (Fig. 3B) shows substantial inter-slide mixing with colors dispersed throughout the UMAP projection. This is further reflected in the UMAP of AdvDINO tile embeddings colored by clusters (Fig. A1), where each cluster contains tiles from multiple slides, suggesting that AdvDINO may be capturing biological patterns that generalize across slides, rather than overfitting to slide-specific artifacts. Quantitatively, AdvDINO achieved a low ARI between cluster assignments and slide IDs of 0.037, compared to a high ARI of 0.66 for DINOv2, confirming that DINOv2 clusters are strongly dictated by slide identity whereas AdvDINO clusters are not.

### 4.2. Proteomic & Prognostic Associations of Clusters

We next characterized associations between the clusters and patient prognosis and proteomic markers. As shown in Fig. 4, clusters were first ranked by C-index, quantifying the concordance between cluster abundance—the proportion of tiles assigned to each cluster per slide—and patient overall survival. Several significant associations were observed, even when adjusting for cancer stage and multiple comparisons (see Methods). Higher abundance of Clusters 2, 5, 13, 23, 25, 26, 28, and 29 were associated with longer survival (C-index of 0.67-0.73), and Cluster 12 was associated with shorter survival (C-index of 0.23).

In addition to prognostic associations, some clusters show distinct proteomic profiles, as quantified by the average pixel intensity for each mIF marker per cluster (Figure 4). Cluster 5, which also shows an association with longer survival, exhibits elevated intensity of DAPI, CD8, and PD-1, suggesting immune cell enrichment. Cluster 1 is enriched for FOXP3, a marker of regulatory T cells that can indicate an immunosuppressive microenvironment, while Cluster 10 shows elevated PD-L1 expression, a checkpoint protein often associated with immune evasion by tumor cells.

While certain survival-associated clusters exhibit distinct marker patterns (e.g., Cluster 5), others are more nonspecific. Cluster 12—the cluster most negatively associated with

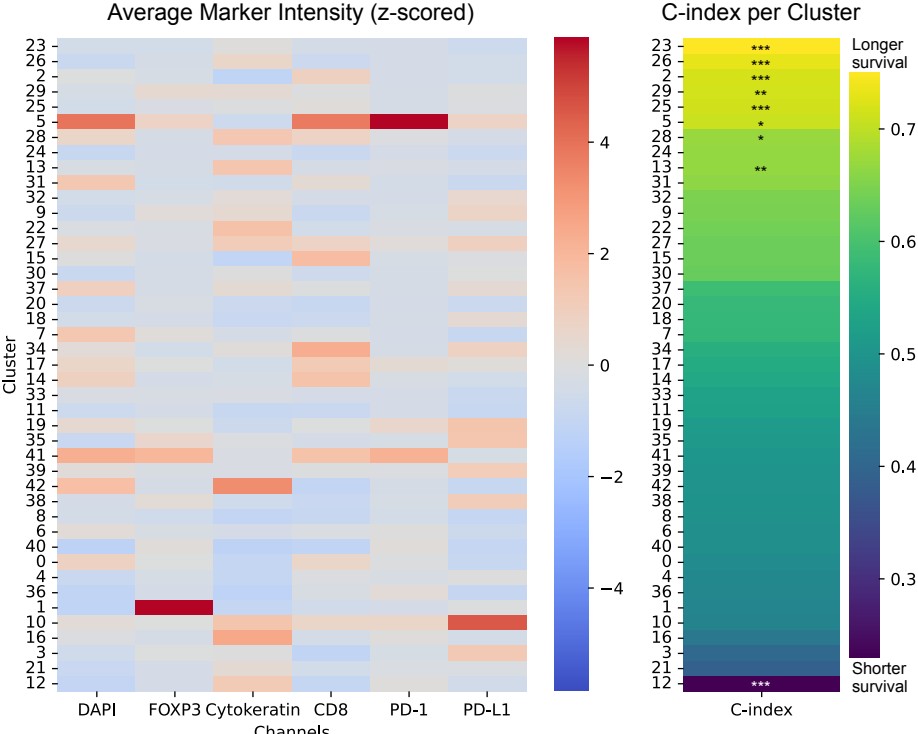

Figure 4: AdvDINO clusters characterized by their z-scored average mIF marker intensity profiles (left) and prognostic relevance (right). * p < 0.05, ** p < 0.005, *** p < 0.0005 (Bonferroni corrected); note: cluster 4 is also significant but is not starred because it does not remain significant when adjusted for cancer stage.

survival—and Cluster 26—the second most favorable cluster—both show slight cytokeratin enrichment and slightly lower intensities of other markers. Visualization of example tiles within these clusters, however, illustrates distinct histological patterns (Figure 5). Cluster 12 tiles tend to contain a sparse set of tumor cells, while Cluster 26 largely represents normal lung tissue. Cluster 23—the most favorable cluster—similarly exhibits normal lung tissue, with a trend towards higher inflammation than Cluster 26. Examples of Cluster 5 confirm the presence of immune enrichment and specifically appear to represent lymphoid aggregates. Examples for the remaining prognostic clusters are contained in Figure A2.

Together, these results show that AdvDINO not only identifies slide-robust clusters, but that these clusters represent distinct biological features with prognostic relevance.

### 4.3. Survival Prediction with ABMIL

Beyond cluster associations, we trained ABMIL models using the AdvDINO embeddings to directly predict patient overall survival. Results are summarized in Table 1. ABMIL using AdvDINO embeddings achieved strong prognostic performance with a C-index of 0.799±0.034 (mean ± SD across folds). This performance remains significant even when adjusting for cancer stage (p<1e-6), suggesting that the model has identified prognostic features not captured in current staging systems. ABMIL using the non-adversarial DINOv2

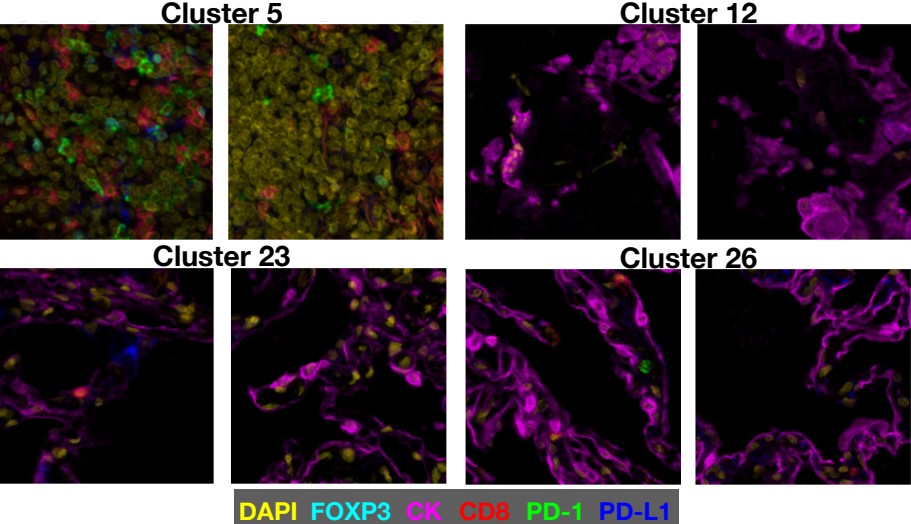

Figure 5: Representative tile images from selected clusters. Clusters 5, 23, and 26 are associated with longer survival; cluster 12 is associated with shorter survival.

Table 1: Weakly supervised prognostic performance.

| Model | C-index Mean $\pm$ SD |
|---|---|
| AdvDINO | **0.799 $\pm$ 0.034** |
| DINOv2 | 0.789 $\pm$ 0.028 |
| Cell Density CoxPH | 0.680 $\pm$ 0.044 |

embeddings exhibits a C-index of 0.789±0.028. Both models outperform a CoxPH baseline using traditional cell density metrics (0.680±0.044), highlighting the potential of state-of-the-art AI methods to more fully leverage spatial proteomic data compared to conventional approaches.

Using the tile-level attention weights generated by ABMIL, we examined associations between the ABMIL-predicted risk scores and the AdvDINO clusters to gain insight into features driving the predictions. All slides (concatenated across folds) were divided into high- and low-risk groups based on the median ABMIL-predicted survival score. For each slide, we selected the top 25 most-attended tiles by ABMIL and calculated the proportion of clusters represented by these tiles within the high-risk and low-risk groups. A difference in cluster proportions was formulated as $\Delta$ = proportion in high-risk group − proportion in low-risk group. As shown in Figure 6, the clusters differentially attended to by the ABMIL model also tend to be those that were found to be the most prognostic in the unsupervised analysis. Cluster 12 is the most over-represented in high-risk predictions, aligning with its ranking as the cluster most negatively associated with survival (C-index < 0.5). In contrast, Cluster 26 is most over-represented in low-risk predictions, aligning with its favorable association with survival (C-index > 0.5). Interestingly, despite low prognostic relevance on its own, Cluster 1 is the second most enriched cluster in high-risk predictions,

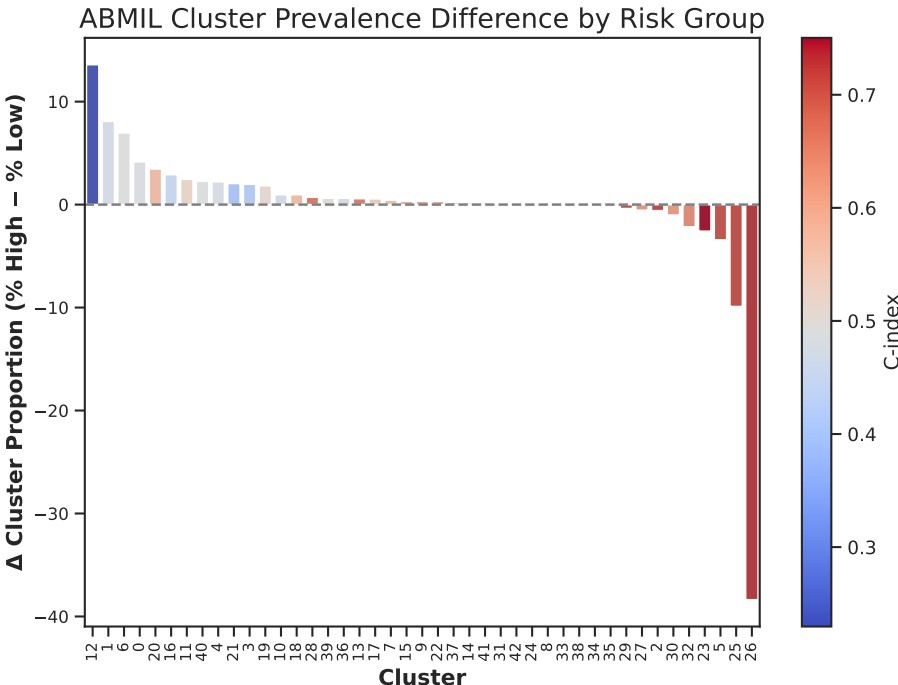

Figure 6: Difference in cluster prevalence amongst ABMIL-attended tiles between high- and low-risk predictions.

aligning with its enrichment of FOXP3 (Figure 4). Conversely, Cluster 5—representing enrichment of CD8 and PD-1 (Figure 4)—is over-represented in low-risk predictions.

## 4.4. Generalizability of Approach to a Breast Cancer Cohort

To assess generalizability, we applied AdvDINO to a separate set of 218 breast cancer mIF WSIs generated using the ImmunoProfile mIF assay (Alessi et al., 2025; Lindsay et al., 2025). Training models on this cohort using the same hyperparameters and approach as the NSCLC cohort, we find that AdvDINO again improves the robustness of the learned representations compared to DINOv2, with an ARI improvement from 0.44 to 0.12. For a downstream application, we evaluated whether the AdvDINO-learned clusters associate with the presence of triple negative breast cancer (TNBC), an aggressive subtype for which differences in the tumor microenvironment remain of significant biological and clinical interest. We find that 8 clusters (out of 49) are significantly associated with TNBC (Bonferroni-corrected and adjusted for cancer stage), compared to 0 for DINOv2. Further details and cluster examples are contained in Appendix A.5. Notably, the cluster most positively associated with TNBC (Cluster 2) shows a high presence of immune cells, consistent with prior reports of immune enrichment and favorable immunotherapy response in TNBC (Schmid et al., 2024; Abdou et al., 2022).

## 5. Conclusion

We introduced AdvDINO, a domain-adversarial SSL framework that learns robust and transferable representations from multi-channel images under domain shift. Applied to six-channel mIF WSIs, AdvDINO substantially reduced slide-specific biases that were apparent in the default implementation of DINOv2. AdvDINO's domain robustness resulted in biologically- and clinically-meaningful clusters, a common goal in spatial proteomics, rather than clusters driven by batch effects. While large-scale public mIF WSI datasets are currently lacking, it will be important in future work to further validate and benchmark the approach against other domain adaptation strategies and across diverse staining protocols, imaging platforms, and populations, where different domain labels could also be used. Future work could additionally explore applying AdvDINO to other medical imaging domains affected by domain shift, such as histopathology and radiology. Given the increasing use of DINOv2 in these settings, the ability of AdvDINO to learn invariant features from unlabeled, multi-source data may similarly enhance the robustness and transferability of learned representations.

## Acknowledgments

W.L. acknowledges funding support from the Ellison Foundation, the Wong Family Award, the Louis B. Mayer Foundation, the National Institute of Biomedical Imaging and Bioengineering award R21EB035247, and the National Library of Medicine award R01LM014775.

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

# Appendix A. Appendix

## A.1. Additional Methodological Details

### A.1.1. Image Preprocessing

The mIF WSIs consisted of 6 biomarker channels. During image acquisition, a background autofluorescence image is also generated, which was used in the following preprocessing steps. First, to identify tissue-containing (foreground) tiles, all seven channels (6 biomarker channels and autofluorescence channel) were downsampled by a factor of 256 and thresholded separately using Otsu's algorithm (Otsu, 1979). A pixel-wise OR operation was then applied across the resulting binary masks to pool foreground regions across channels. Tiles of fixed size ($256\times256$) were extracted from the detected foreground regions. Background subtraction was then applied to each biomarker channel using the autofluorescence signal. We note that background subtraction is loosely akin to normalizing by overall staining intensity in H&E slides. Next, per-channel intensity histograms were computed across the dataset, which revealed highly skewed distributions toward the lower end. Based on this observation, we clipped pixel intensities to the range [0, 200] and linearly rescaled values to [0, 255]. Finally, each channel was normalized by subtracting the dataset-wide mean and dividing by the dataset-wide standard deviation, a common strategy when creating vision model inputs.

### A.1.2. Unsupervised Clustering

A representative reference set was constructed by randomly subsampling 20% of the tile embeddings ($\approx$1.09 million) across the cohort to enable efficient clustering. Leiden clustering was then performed on this subset. This clustering consisted of several steps. First, principal component analysis (PCA) reduced the embeddings from 1024 to 128 dimensions. A k-nearest neighbors (KNN) graph was then constructed by connecting each tile to its 250 most similar tiles based on Euclidean distance in the PCA-reduced embedding space. This relatively large number of neighbors was chosen to preserve global structure and promote long-range interactions between phenotypically similar tiles across WSIs. We then applied Leiden clustering to this graph with a resolution parameter of 2.0, which encourages the identification of fine-grained clusters. After fitting the clusters on the sampled subset, the cluster labels were propagated to the remaining $\approx$4.37 million tiles based on nearest-neighbor relationships with the reference set.

### A.1.3. Survival Prediction with ABMIL

In multiple instance learning, each WSI is treated as a bag of tile embeddings (e.g., generated by the pretrained AdvDINO encoder). With ABMIL, an attention pooling mechanism is trained to assign a weight to each tile to yield a weighted slide-level representation, enabling patient survival prediction without requiring tile-level annotations. Each slide is labeled with the corresponding patient's follow-up time and survival status (deceased or censored), which serve as weak supervision signals during training. The resulting slide-level representation is passed through a linear layer to predict a vector of logits corresponding to discrete survival intervals, for which we use four time bins defined by the quartiles of observed event times in the dataset. The model is trained using a log-likelihood loss that

accounts for censored survival data by modeling discrete hazard and survival functions (Zadeh and Schmid, 2021). At inference, the predictions across time bins are converted into a cumulative risk score as the sum of the survival probabilities across bins.

### A.1.4. AdvDINO Pretraining Hyperparameters

We adopt the data augmentation scales recommended in Chen et al. (2024), using a global crop scale of [0.48, 1] and a local crop scale of [0.16, 0.48]. We train the model for 430,000 iterations (approximately 10 epochs) using the AdamW optimizer (Loshchilov and Hutter, 2017) with a cosine learning rate schedule and linear warmup. Training uses a batch size of 64 per GPU, yielding an effective batch size of 128 across two NVIDIA A100 GPUs (each with 80GB of VRAM). To improve numerical stability in the KoLeo loss, we increase the hyperparameter $\epsilon$ from $1 \times 10^{-8}$ to $1 \times 10^{-4}$, preventing infinite loss values during training. All other hyperparameters, including the base learning rate, follow the official DINOv2 configuration (Oquab et al., 2024).

### A.1.5. ABMIL Cross-Validation and Hyperparameters

We conducted five-fold cross-validation, stratified by patient survival status (deceased or censored), to evaluate ABMIL performance using the AdvDINO embeddings. In each split, three folds were used for training, one for validation/model selection, and one for testing. The ABMIL models were trained using the AdamW optimizer (Loshchilov and Hutter, 2017) without a learning rate scheduler for 80 epochs, which was sufficient for convergence. Training was conducted on a single NVIDIA A100 GPU with 80GB of VRAM. Three learning rates were considered: $1 \times 10^{-6}$, $1 \times 10^{-5}$, and $1 \times 10^{-4}$, with final selection based on the highest mean C-index across validation folds ($1 \times 10^{-4}$). Using this setting, we report the C-index mean and standard deviation across the five test folds.

## A.2. UMAP and Clustering Implementation

To generate the latent space visualizations, we followed a standardized pipeline using the Scanpy library. First, we performed dimensionality reduction by computing Principal Component Analysis (PCA) on the tile embeddings, retaining the top 128 principal components, thereby preserving variance. We then constructed a neighborhood graph using n=250 neighbors. This higher value was chosen to capture the global manifold structure, ensuring that the resulting UMAP layout accurately reflects the underlying data distribution rather than localized noise. Community detection was then performed using the Leiden algorithm with a resolution of 2.0 and 2 iterations to identify granular sub-clusters within the tile embedding space. The final 2D visualization was generated with UMAP, initialized on the Leiden-associated neighborhood graph using the standard Scanpy defaults (min_dist=0.5, spread=1.0). This procedure ensures that the visual separation of clusters is a result of the learned domain-invariant representations rather than an artifact of hyperparameter tuning.

### A.3. UMAP by Cluster Label

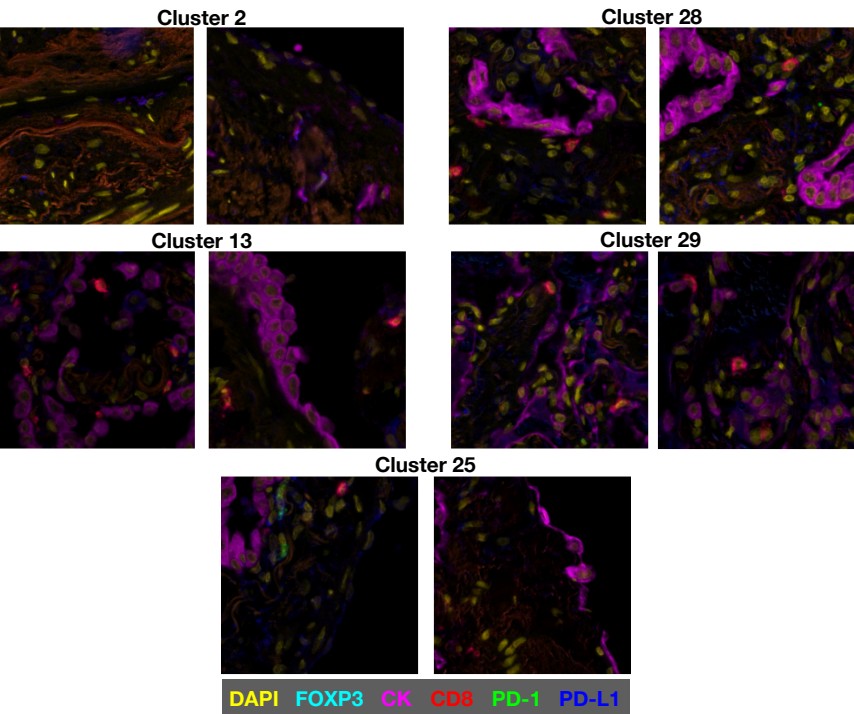

UMAP of Tile Embeddings Colored by Cluster

**A** DINOv2 (non-adversarial)

**B** AdvDINO

Figure A1: (A) UMAP of tile embeddings from the baseline DINOv2 model, colored by unsupervised cluster labels. (B) UMAP of AdvDINO tile embeddings colored by unsupervised cluster labels.

### A.4. Additional NSCLC Cluster Examples

Cluster 2

Cluster 28

Cluster 13

Cluster 29

Cluster 25

DAPI   FOXP3   CK   CD8   PD-1   PD-L1

Figure A2: Representative tile images from remaining prognostic clusters from the NSCLC cohort.

### A.5. Breast Cancer Analysis Details

The breast cancer cohort consisted of 218 primary-site tumor samples from 200 patients, with an mIF WSI generated for each sample as part of the prospective ImmunoProfile cohort at the Dana-Farber Cancer Institute (Alessi et al., 2025; Lindsay et al., 2025). The breakdown by cancer stage was 25% (54 cases) high-stage and 75% (164 cases) low-stage. By subtype, 21% (45 cases) were TNBC, 68% (148 cases) were HR+/HER2-, 6% (13 cases) were HR+/HER2+, 4% (8 cases) were HR-/HER2+, and 2% (4 cases) were unknown subtype. Overall survival data was also curated, with 84% of being censored at the time of last follow up. Given this high censorship rate and the clinical significance of TNBC, we focused on assessing associations between the AdvDINO-learned clusters and TNBC presence (i.e, TNBC given a label of 1, other subtypes given a label of 0). The four tumors with unknown subtype were excluded from the association analysis, but were included in the initial AdvDINO training. Associations were quantified via AUROC with the DeLong method used to compute p-values, which were Bonferroni corrected. Similar to the NSCLC analysis, we retained only clusters that were significant when also adjusting for cancer stage, which was done via a logistic regression with the cluster proportions and stage as covariates. Examples of the most positively and negatively TNBC-associated clusters are shown in Fig. A3.

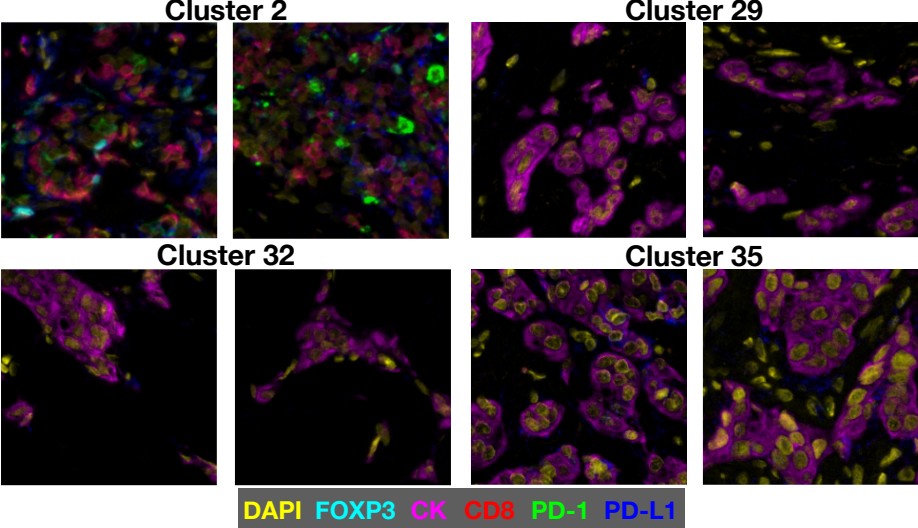

Figure A3: Representative tile images from clusters most associated with TNBC in the breast cancer cohort. Cluster 2 is positively associated with TNBC (AUROC of 0.68, p=0.01, Bonferroni corrected). Clusters 29, 32, and 35 are negatively associated with TNBC (AUROC of 0.23, 0.25, 0.25, respectively; p < 0.0001).

