# OpenReview forum: "AdvDINO: Domain-Adversarial Self-Supervised Representation Learning for Spatial Proteomics"
_MIDL.io/2026/Conference — MIDL 2026 Poster_

### Official Review · Reviewer_ki4C · 2026-01-01

**Confidence:** 5
**Preliminary Rating:** 5
**Final Rating:** 5

**Summary:**

The authors propose an SSL-based solution to support the use of whole-slide images (WSIs) for analyzing the tissue composition, spatial properties and microenvironment of tumors, and predicting patient outcomes in oncology. Their solution is built upon the well-known DINOv2 pipeline and training recipe, plus an existing technique for adversarial learning tweaked for robustness to domain-shift.

**Strengths:**

While technical novelty is limited, the paper is well-written and demonstrates a powerful application for medical imaging with high potential for extending to other imaging-based specialties of medicine. It is the most mature and striking paper in my batch. I thank the authors for their meticulous work.

**Weaknesses:**

Regarding generalization, it would be good to test the models trained on a particular set of WSIs on an external collection. In Section 4.4, it is said that the overall AdvDINO training recipe was applied on a breast cancer dataset from scratch (or UNI initialization). While this demonstrates the applicability of the approach in different clinical scenarios, generalization to different collections within the same clinical setup is also important. In this respect, what happens if you transfer the models trained on NSCLC to TNBC without retraining? Does it perform comparably in a linear probing scenario? Are these datasets totally different in terms of imaging characteristics? Why did you retrain the model on TNBC? If these are clarified properly, new experiments may not be needed, but we need to understand the generalization game here better.

I have other suggestions for refinement; however, I don’t consider them as major weaknesses at this point. So, I will list them in the detailed comments below.

**Detailed Comments:**

Page 3: Domain Invariant Representation Learning:
What is SASSL? Could you spell out the full form?

Page 6: Evaluation Metrics and Statistical Analysis:
What is ARI? How does it work? A brief explanation could help the reader understand it better. You did it already for C-index on page 5. ARI would benefit from it, too.

UMAP results and visualizations: While the overall message is understood, it would be nice to see more details about the UMAP settings. I don’t want to delve into UMAP vs tSNE comparisons but we can collapse or enlarge clusters through hyperparameters to obtain similar visual effects, if we spend some time to that end. To improve clarity, could you describe your UMAP procedure in more detail? This can be done in Appendix.

**Justification Of Final Rating:**

The manuscript was already in a decent form prior to rebuttal. The authors also addressed my comments well. Given the additional comments from other reviewers, I believe this is a good to work to appear at MIDL 2026. To give it an edge in this respect, I am updating my preliminary rating from "weak accept" to "strong accept".

An extended version for MELBA Special Issue would be also nice for exploring the potential of AdvDINO in greater detail.

**Justification Of The Preliminary Rating:**

In times where everybody talks about large models and tries to scale things up, the authors identified a good problem and studied it well. I mainly evaluated the work from a technical perspective since I am not an expert in proteomics or oncology. I hope they receive a proper evaluation from the clinical side, as well. I will be happy to adjust my rating based on the rebuttal outcomes.

**Questions To Address In The Rebuttal:**

Those in the detailed comments should be doable during the rebuttal period.

---

> ### Author Response · Authors · 2026-01-25
>
> We thank the Reviewer for their constructive comments and positive feedback. We respond to specific comments below:
>
> **Breast cancer motivation/generalization**: Thank you for the opportunity to clarify. Our intent was to mimic common spatial proteomics workflows, where investigators often perform the assay on a particular disease (e.g. cancer type) to better understand the biology of the disease. A common goal of this analysis is to identify different disease phenotypes (e.g. clusters) associated with clinically-important variables, such as survival outcomes or molecular subtypes. These phenotypes are typically derived from hand-engineered metrics (e.g. counts of different cell types), but our results suggest that they can instead be identified in a data-driven, self-supervised fashion directly from mIF images using AdvDINO. Importantly, these findings hold both for a relatively large NSCLC cohort (435 samples) as well as a moderately-sized breast cancer cohort (218 samples). Furthermore, based on Reviewer 1’s request, we have assessed whether the DINOv2 clusters are associated with triple negative breast cancer (TNBC) status in the breast cancer cohort. We find that none of the DINOv2 clusters are significantly associated with TNBC, and thus this finding is exclusive to AdvDINO. We will expand upon these motivations in the final version, as well as emphasize the importance of future assessments of generalization, which we envision can be supported by our public code.
>
> **SASSL**: Thanks for pointing this out. SASSL stands for “Style Augmentations for Self Supervised Learning”, which we will spell out in the final version.
>
> **ARI**: The Adjusted Rand Index (ARI) is a metric for comparing two clusterings of the same data. It measures the concordance between the clusterings while adjusting for chance, where a value of 0 indicates chance concordance and 1 indicates perfect agreement. In our case, each image tile is associated with a WSI label (indicating the slide it came from) and a cluster label derived from model embeddings. If each of the embedding clusters consisted almost entirely of tiles from a single WSI, then the ARI would be high and it would suggest that the clusters reflect slide-specific batch effects rather than underlying biological phenotypes. We will include this explanation in the updated version.
>
> **UMAP**: We agree that UMAP/tSNE visualizations are often over interpreted, which is why we were keen on including ARI as a quantitative metric that is independent of visualizations. We will include the following paragraph below in the Appendix for the final version, which specifies the PCA preprocessing steps, number of neighbors, and clustering resolution used to generate the UMAPs. Briefly, we used 128 principal components to retain the majority of the variance, n=250 neighbors to capture the global structure, and a resolution of 2.0 for Leiden clustering to ensure fine-grained community detection:
>
> "To generate the latent space visualizations, we followed a standardized pipeline using the Scanpy library. First, we performed dimensionality reduction by computing Principal Component Analysis (PCA) on the tile embeddings, retaining the top 128 principal components, thereby preserving variance. We then constructed a neighborhood graph using n=250 neighbors. This higher value was chosen to capture the global manifold structure, ensuring that the resulting UMAP layout accurately reflects the underlying data distribution rather than localized noise. Community detection was then performed using the Leiden algorithm with a resolution of 2.0 and 2 iterations to identify granular sub-clusters within the tile embedding space. The final 2D visualization was generated with UMAP, initialized on the Leiden-associated neighborhood graph using the standard Scanpy defaults (min_dist=0.5, spread=1.0)."

---

### Official Review · Reviewer_U1MY · 2026-01-09

**Confidence:** 4
**Preliminary Rating:** 4
**Final Rating:** 5

**Summary:**

Su et al., introduce a new self-supervised learning approach that integrates an adversarial learning component (a Domain Adversarial NN) into DINOv2 (AdvDINO) to overcome well-known domain shift challenges in biomedical imaging (e.g., differences in batch labelings, slices or scanners). The Domain Adversarial NN is connected to the Student ViT of DINOv2 and trained to infer the ID of the experimental domain shift source (e.g., batch, slice), making the Student ViT more agnostic to imaging experimental variations. Focusing on whole-slide tumoral tissue proteomics image data and tile-based image processing for prognosis purposes, the authors show improved tile representation capabilities of AdvDINO—without minimal domain shift bias—compared DINOv2, and that these tile representations are coherent with patients' survival prognosis.

**Strengths:**

- Self-supervised approaches, particularly self-distillation, are powerful techniques to synthesise the tones of information available in the emerging multiomics techniques, crucial for better understanding tissue microenvironments, in physiology and disease. On the other hand, domain shifts and experimental variability are a major bottleneck for the use of data-driven approaches with biomedical techniques such as digital pathology, tissue multiomics, and overall imaging techniques involving tissue staining and patients' samples, among many others.  While there exist some more basic solutions such as as patient&experiment-aware data normalisation, these often do not work when there is a lack of control or sufficient knowledge about the image acquisition process. Therefore, more systematic and generalisable approaches capable of overcoming this issue are very much needed. Here, Su et al., propose an improved methodology in this direction.
- AdvDINO manages to better discard the implicit experimental bias (shown in Fig 2)
- To further show the usability and medical relevance of AdvDINO representation, the authors perform a tile clustering that correlates with high/low patient survival risk and achieves a relatively high prognostic performance (0.799 and 0.789 for DINOv2) compared to the baseline (0.689) (see weaknesses).
- The manuscript is well written, being very easily understandable
- The authors have made their code publicly available

**Weaknesses:**

- Lack of comparison with other methods integrating domain adaptation. This would highlight both novelty and any potential improvement of the authors' proposal.
- The authors show that DINOv2' representations are affected by domain shift biases, however, to my understanding in this paper, DINOv2 was not retrained. At the same time, the accuracy values shown in Table 1, are pragmatically similar, which brings the question of how useful or what is the real gain of AdvDINO against DINOv2. Moreover, in Fig 3 a) a slice-wise bias is clear, but there is also some in Fig 3 b), where slices tend to be close to each other, and in some cases, form clear clusters. To some extent, as the slices represent a specific tumor microenvironment and patient, it is expected that they tend to aggregate with each other, as those are strong semantic priors for the corresponding tiles. The second question that rises then, is whether the important bias comming from staining protocols for example, and the one that prevents a more general tumor phenotyping, has been indeed removed in AdvDINO or is it to some extent, still present.
Potentially and if time allows, the authors could consider (1) running and comparing the same clustering and progonostic analysis with the outputs from DINOv2, to identify potential gainings, (2) showing the potential of a pretrained AdvDINO with new data (e.g., the one for breast cancer), or (3) find a more clear example of AdvDINO advantage.

**Detailed Comments:**

- Figure 1: Are these slices from the same patient? The slices show a strong variability but in the figure, as it is, it is not clear if the variability is medically relevant. Potentially, the authors could show this variability for slices ranked with the same patient's survival risk, or for tiles that belong to the same cluster but not to the same slice. I would also suggest adding a scale bar and zooming in on the tiles to enable marker distinction.

- Student ViT: Is the output of this network updated to connect with the Domain Adversarial NN, or is it kept the same as in DINOv2. I would suggest being more explicit about it in the text.

- Figure 4: Please, turn around a and b to start reading the legend according to the figure from the left to the right. Also, in the legend of that figure, "note cluster 4", does it refer to cluster 24?

- Figures 5, A2 and A4: It would be easier to understand the differences if a C score or "longer/shorter" survival legend accompanies each cluster title in the figure.

- While references to the data are given in the text and it seems to be publicly available, in the corresponding manuscripts I only found the following link to access the images (https://www.synapse.org/Synapse:syn52596661/wiki/634052); however, I found no images there. Is that the correct one? Would it be possible to directly provide a link to the data?

**Justification Of Final Rating:**

Most comments were addressed in the review, particularly the gain of AdvDINO against DINOv2, improving the major concerns about the work. This is meant to be updated in the new version of the manuscript, as stated:
"We find that none of the DINOv2 clusters are significantly associated with TNBC, and thus this finding is exclusive to AdvDINO. We will include these results and clarifications in the final version. We will also explicitly acknowledge the importance of further benchmarking and comparing to other methods in future work."
Overall, this is an interesting contribution to the field and presents an important highlight regarding domain shifts and self-supervised approaches.

**Justification Of The Preliminary Rating:**

The authors propose a methodological approach to overcome a known bottleneck in biomedical imaging. The manuscript is clearly written, with a nice visualisation of results and demonstration of the potential of self-supervised learning for prognosis. However, it lacks further validation of actually solving the described problem and the impact of the new method. I would agree that these points can be addressed by a clearer discussion, explicit statements or potentially, quick addition of results.

**Questions To Address In The Rebuttal:**

The questions are detailed in the weaknesses and detailed comments section

---

> ### Author Response · Authors · 2026-01-25
>
> We thank the Reviewer for their constructive comments and positive feedback. We respond to specific comments below:
>
> **DINOv2 training and AdvDINO advantages**: Thank you for the opportunity to clarify. We did retrain the DINOv2 model. It was trained on the mIF data in the same fashion as AdvDINO, except without the domain discriminator. As DINOv2 is a leading self-supervised learning algorithm in medical imaging, our findings that the default implementation results in representations that are heavily influenced by domain shift, but that these effects are mitigated by AdvDINO, have important implications for the field. The domain robustness of AdvDINO allows us to identify biologically meaningful clusters across the cohort, a common goal in spatial proteomics, rather than clusters driven by batch effects. As further evidence of this benefit, based on Reviewer 1’s request, we have assessed whether the DINOv2 clusters are associated with triple negative breast cancer (TNBC) status in the breast cancer cohort. We find that none of the DINOv2 clusters are significantly associated with TNBC, and thus this finding is exclusive to AdvDINO. We will include these results and clarifications in the final version. We will also explicitly acknowledge the importance of further benchmarking and comparing to other methods in future work.
>
> **Figures**: Thank you for these helpful suggestions. We will be sure to reorder the panels in Figure 4 and incorporate the suggestions for Figure 5 in the final version. For Fig. 4, we confirm that the legend is intended to reference cluster 4. In Figure 1, the slides are from different patients but were chosen based on similar overall morphology.
>
> **Student ViT**: Thank you for the opportunity to clarify. Yes, the output of the Student ViT serves as input to the Domain Adversarial Neural Network, in addition to being used for the self-supervised learning objectives. The internal architecture (the transformer blocks and attention mechanisms) of the Student ViT otherwise remains the same as the original DINOv2 backbone. This process allows the student’s weights to be optimized for both self-supervised and adversarial objectives without architectural modifications. We will include this clarification in the final version.
>
> **Imaging availability**: Unfortunately the images have not yet been IRB-approved for release. We agree that they would provide a useful resource, which we intend to pursue.

---

### Official Review · Reviewer_iQmm · 2026-01-10

**Confidence:** 4
**Preliminary Rating:** 4
**Final Rating:** 4

**Summary:**

AdvDINO integrates a gradient reversal layer into DINOv2 to learn domain-invariant representations from multiplex immunofluorescene (mIF) whole slide images. The motivation is that standard SSL methods capture slide-specific batch effects rather than true biological variations. In other words, the learned embeddings tend to group tiles by slides of origin potentially due to slide-consistent staining or imaging artifacts and this happens more in SSL than supervised learning (as there is no task forcing the model to ignore nuisance variation). The proposed method is trained on 435 lung cancer mIF WSIs. It is shown to reduce slide-level clustering compared to vanilla DINOv2 and yield clusters with distinct (a) proteomic profiles and (b) prognostic associations. Survival prediction via ABMIL and generalizability on a breast cancer cohort is also shown (where clusters associate with TNBC status).

**Strengths:**

- The problem addressed in real and under-addressed. Batch effects in mIF are substantial and most SSL works ignores this. Figure 1 shows this issue clearly.
- Dataset scale is respectable for mIF (full WSIs rather than TMAs) - 435 slides and proper clinical follow-up. This is harder to obtain / curate than standard H&E datasets.
- Multiple evaluation axes strengthen the claims: 1. domain robustness via ARI, 2. biological validity via marker profiles, 3. clinical utility via survival prediction; not just benchmark SOTAs.
- The biological findings are sensible. For example: cluster 5 showing CD8/PD-1 enrichment correlating with survival aligns with known immunology and cluster 12 with sparse tumor cells predicting worse outcomes makes clinical sense.
- Extension to breast cancer with TNBC associations shows this is not overfit to one cohort.

**Weaknesses:**

- Methodological novelty is limited. GRL in DANN is not new (Ganin et al., 2016). The contribution is primarily in application to mIF rather than algorithm.
- Baseline comparisons are limited to vanilla DINOv2. Including at least one alternative robustness strategy (e.g. stain normalization or style transfer-based augmentation) would strengthen claims.
- Discriminator uses a 435-way classification (one class per slide). This is unusual for adversarial domain adaptation which typically uses few domains. What is the discriminator accuracy? It would help interpret how adversarial signal behaves in extreme classification settings.
- ABMIL improvement is not convincing. The C-index overlap substantially. No statistical test is provided for this comparison. The claim that AdvDINO improves survival prediction needs more support.
- Lambda_adv=50 selection is hand-wavy. Authors say it was chosen to balance losses "because DINOv2's high computational costs limit extensive hyperparameter optimization." This is understandable but an ablation with 2-3 values would help.
- All data comes from one institution with one assay (which the authors acknowledge in the limitation discussion). True domain shift would involve different centers, scanners or staining protocols. Using slide ID as domain label only addresses within-cohort variation. Likewise, the breast cancer validation uses the same assay and institution. It shows the method transfers to another cancer type but not to truly different domains.

**Detailed Comments:**

- Both "DINOv2 baseline" and AdvDINO start from UNI, both finetune on mIF. Only difference is the adversarial loss. Showing frozen UNI features also have batch effects on mIF would strengthen motivation (to localize whether batch effects emerge during SSL finetuning or earlier).
- Channel count inconsistency. Abstract says six channels, but Section A.1.1 mentions seven grayscale channels (six biomarkers plus autofluorescence). The model input is stated as 6-channel after background subtraction using autofluorescence. This should be clarified upfront.
- Figure 4 asterisk notation for Cluster 4 is confusing. Caption says significant but not starred because it doesn't survive stage adjustment. Consider cleaner notation.
- The Leiden clustering uses resolution=2.0 yielding 43 clusters. Different resolutions would give different cluster counts. Are the prognostic findings robust to this choice?
- For the breast cancer cohort, you report ARI improvement but no comparison of downstream TNBC associations between AdvDINO and DINOv2. Does DINOv2 also find TNBC-associated clusters?

**Justification Of Final Rating:**

The new TNBC analysis is convincing. DINOv2 finds no significant cluster associations while AdvDINO does. This answers my main question about whether the adversarial component adds real value beyond standard SSL. It does. The authors also handled the ABMIL result transparently by acknowledging the improvement is not statistically significant and committing to remove overclaims.

My remaining concern is the discriminator dynamics. The authors compare their 435-class setup to ImageNet classification, but this analogy does not hold. ImageNet classes have semantic structure; slide IDs are arbitrary labels. Adversarial domain discrimination behaves differently from standard classification, and in extreme multi-class settings the gradient signal may saturate or become noisy. Without reporting discriminator accuracy, we cannot tell if the adversarial loss is providing meaningful supervision or operating near chance. This is not a fatal flaw but it leaves a gap in understanding why the method works.

Alternative baselines (stain normalization, other domain adaptation methods) are still missing. The mIF sparsity argument for skipping stain normalization is reasonable but not fully convincing.

Overall the paper solves a real problem with demonstrated utility on two cohorts. The empirical results are solid even if mechanistic understanding is incomplete. Suitable for MIDL.

**Justification Of The Preliminary Rating:**

This work addresses a genuine and under-addressed problem in spatial proteomics. Batch effects in mIF imaging (obscuring biological signals) are real and the motivation is clearly shown. The solution is technically sound. Experiments cover relevant axes including biological interpretation and downstream clinical utility. While the method is incremental, application papers with solid validation are appropriate for MIDL. Main concerns are the (a) lack of alternative baselines, (b) unconvincing ABMIL improvement and (c) single institution data limiting true domain shift claims (which the authors acknowledge). Overall, the work provides a practical pathway for robust SSL in spatial proteomics and I believe the contribution is sufficient for MIDL given the problem importance and experimental thoroughness within scope.

**Questions To Address In The Rebuttal:**

- What is the domain discriminator's classification accuracy during training and at convergence? With 435 classes, how does the adversarial learning dynamics behave?
- Can you provide a paired statistical test for ABMIL improvement (AdvDINO vs DINOv2) across the five CV folds?
- Why no comparison with stain normalization baselines? Even simple intensity standardization would contextualize the adversarial approach.
- How sensitive are the prognostic cluster findings to Leiden resolution parameter?
- For breast cancer, do DINOv2 clusters also associate with TNBC, or is this specific to AdvDINO?

---

> ### Author Response · Authors · 2026-01-25
>
> We thank the Reviewer for their constructive comments and the statement that the contribution is sufficient for MIDL. We respond to specific comments below:
>
> **DINOv2 clusters for breast cancer**: Thank you for this suggestion. We have now performed this analysis and have found that none of the DINOv2 clusters are significantly associated with TNBC, and thus the results are specific to AdvDINO. We will include this result in the final version.
>
> **ABMIL statistical test**: We agree that both the DINOv2 and AdvDINO models perform strongly under ABMIL. Based on the Reviewer’s suggestion, we have performed a paired t-test over the five CV folds and the results are not significant. We were intentionally cautious when describing this comparison originally, and will ensure that there are no mentions of outperformance in the final version, along with including this statistical result. Importantly, this result is not at odds with the feature/cluster analysis, where AdvDINO learns more domain-robust representations, even if a supervised model (ABMIL) on top of DINOv2 embeddings can also predict prognosis. This enhanced robustness is critical for spatial proteomics applications, as a common goal is to identify different disease phenotypes (e.g. clusters) associated with clinically-important variables. These phenotypes are often derived from hand-engineered metrics (e.g. counts of different cell types) but our results suggest that they can instead be identified in a data-driven, self-supervised fashion directly from mIF images using AdvDINO (but not DINOv2). This utility is further supported by the Reviewer’s request above, where cluster association with TNBC status was exclusive to AdvDINO.
>
> **Stain normalization/Additional baselines**: Thank you for the comment. While stain normalization is a natural approach for H&E images, it is challenging to adapt to mIF given the multi-channel nature and the inherent sparsity of mIF images (many pixels are not positive for any biomarker and any given biomarker may only be present in a subset of images). We did however perform background subtraction using autofluorescence during preprocessing (Appendix A.1.1), which is loosely akin to normalizing by overall staining intensity in H&E. Importantly, this was insufficient to remove batch effects with DINOv2 (e.g. Fig 3). Nonetheless, we will expand the limitations in the final version to acknowledge the utility of benchmarking against other domain adaption strategies.
>
> **Domain discriminator dynamics**: We did not explicitly log classification accuracy during training and training a new model is infeasible in the timeframe, but we note that there are approximately 12K training examples per class (5.46M tiles / 435 WSIs), providing many examples for domain learning. As a comparison, ImageNet contains 1000 classes and ~1.3K images per class. Thus, even though 435 classes may be larger than typical domain discriminator applications, it is a reasonable size for AI classification tasks in general and our results support the effectiveness of the adversarial approach in mitigating batch effects in the learned representations (e.g. Figure 3).
>
> **Channel count inconsistency**: Thank you for the opportunity to clarify. There are 6 biomarkers in the assay, resulting in 6 mIF image channels. During image acquisition, a background autofluorescence image is also generated, which is concatenated to the remaining channels in the resulting data structure. Our models operate on the 6-channel mIF images, where we first use the autofluorescence image for background subtraction. We will clarify this in the final version.
>
>
> **Leiden resolution**: While there is insufficient time to perform clustering over the cohort, we did so on one CV fold (82 slides) using a lower Leiden resolution (0.8 instead of 2.0) to explore the potential impact of fewer clusters. This resulted in 19 clusters (compared to 43 originally) and even with this reduced subset, 3 out of the 19 clusters were significantly associated with prognosis (p<0.05, BF-corrected).

---

### Comment · Area_Chair_MgQt · 2026-01-30
**Reviewers, please take a moment to review the rebuttals and revise your ratings**

Update your final rating by clicking “Edit” → “Official Review” and providing the Final Rating by February 1st 2026 (23:59 AoE).

---

> ### Comment · Reviewer_U1MY · 2026-01-30
>
> Unfortunately, I cannot see the final version of the manuscript nor the track of changes. Could you please, let me know where I can find it?

---

> > ### Comment · Area_Chair_MgQt · 2026-01-30
> >
> > Thank you for asking. It appears the authors have chosen not to update their manuscript during this phase of review. Note that per instructions from the Program Chairs, "a revised PDF file is optional and no PDF file should NOT be a defect or disadvantage when giving your rating." Please evaluate the quality of the authors’ intended responses based on their written rebuttal. If you find that the rebuttal is inadequate or does not give you confidence that your original concerns would be sufficiently addressed, feel free to state this and reflect it in your evaluation. I will take your assessment into account when making my final recommendation.

---

### Meta-Review · Area_Chair_MgQt · 2026-02-09

**Recommendation:** Accept (Oral)
**Confidence:** 5

**Metareview:**

The authors present a domain-adversarial self-supervised learning framework, AdvDINO, that integrates a gradient reversal layer into DINOv2 and evaluates its ability to represent spatial proteomic data and to learn biologically relevant features to predict patient survival. Reviewers found that the work addresses an important problem related to robustness against domain shifts, that multiple experiments provide complementary views of the framework's performance, and that the manuscript is clearly presented. The authors were found to be largely responsive to the reviewers' critiques, providing additional statistical testing and clarification. While one reviewer noted the lack of reporting on discriminator accuracy, reviewers were largely enthusiastic about the work.

---

### Decision · Program_Chairs · 2026-02-13

Accept (Poster)